# Pangea: An MLOps Tool for Automatically Generating Infrastructure and Deploying Analytic Pipelines in Edge, Fog and Cloud Layers

**DOI:** 10.3390/s22124425

**Published:** 2022-06-11

**Authors:** Raúl Miñón, Josu Diaz-de-Arcaya, Ana I. Torre-Bastida, Philipp Hartlieb

**Affiliations:** 1Digital, TECNALIA, Basque Research and Technology Alliance (BRTA), Parque Tecnológico de Álava Albert Einstein 28, Vitoria-Gasteiz, 01510 Álava, Spain; josu.diazdearcaya@tecnalia.com (J.D.-d.-A.); isabel.torre@tecnalia.com (A.I.T.-B.); 2Mining Engineering and Mineral Economics, Montanuniversitaet Leoben, Erzherzog-Johann-Straße 3, 8700 Leoben, Austria; philipp.hartlieb@unileoben.ac.at

**Keywords:** edge, cloud, analytic pipeline, MLOps, infrastructure, mine

## Abstract

Development and operations (DevOps), artificial intelligence (AI), big data and edge–fog–cloud are disruptive technologies that may produce a radical transformation of the industry. Nevertheless, there are still major challenges to efficiently applying them in order to optimise productivity. Some of them are addressed in this article, concretely, with respect to the adequate management of information technology (IT) infrastructures for automated analysis processes in critical fields such as the mining industry. In this area, this paper presents a tool called Pangea aimed at automatically generating suitable execution environments for deploying analytic pipelines. These pipelines are decomposed into various steps to execute each one in the most suitable environment (edge, fog, cloud or on-premise) minimising latency and optimising the use of both hardware and software resources. Pangea is focused in three distinct objectives: (1) generating the required infrastructure if it does not previously exist; (2) provisioning it with the necessary requirements to run the pipelines (i.e., configuring each host operative system and software, install dependencies and download the code to execute); and (3) deploying the pipelines. In order to facilitate the use of the architecture, a representational state transfer application programming interface (REST API) is defined to interact with it. Therefore, in turn, a web client is proposed. Finally, it is worth noting that in addition to the production mode, a local development environment can be generated for testing and benchmarking purposes.

## 1. Introduction

### 1.1. Problem Statement

In recent years, machine learning (ML) has been considered one of the technologies driving the increasing value of companies, treating data as an important asset to make decisions in a more agile, coherent and sometimes automatic way. This has been fuelled by the introduction of big data technologies to enable the development of new algorithms at scale [1] and by increased data generation in the last decade. Jones et al. [2] estimated that there are seven billion connected devices in 2022 and that the number will increase to 22 billion by 2025. In addition, the more generalised use and reduction in cost of specific hardware technologies such as graphics processing units (GPUs) will establish a promising environment for motivating companies to experiment with artificial intelligence (AI) technologies.

However, to fully take advantage of the claimed benefits of ML, more than just the model building phase should be considered. There are still several aspects in the machine learning life cycle that must be addressed. For instance, data scientists create mathematical or machine learning models to make machines more intelligent by using a wide range of methods, techniques and algorithms such as regression, classification, optimisation or clustering. They excel in the creation of such models. However, as highlighted in [3], they usually do not have the software engineering skills to deploy their code in real production environments. This is because they usually have a strong mathematical background. Nevertheless, the skills required are rather different, involving, among others, a detailed knowledge of distinct environments, file formats, protocols and networks, being able to provide a fault-tolerant scalable and distributed application or using specific techniques to enhance and accelerate the software development life cycle.

Behind this idea, the machine learning operations (MLOps) paradigm has emerged. As stated by Alla and Adari (2021) in [4], MLOps could be understood as an intersection between machine learning and DevOps techniques. DevOps [5] is considered to be a set of practices aiming to improve the software life cycle guaranteeing the continuous delivery. In the same way, MLOps is focused on applying the DevOps perspective to manage high-performance machine learning models, enabling continuous delivery to enhance their life cycle, and, in turn, easing the duties of data analysts and data engineers.

One of the key challenges of MLOps is the rapid deployment of machine learning models in production environments. In the book of Cloudera [6], they propose to address the following challenges to accomplish this MLOps process:1.Model packaging must be considered to enable automatising the ML life cycle where an enormous tool ecosystem is offered.2.Model deploying involves providing the model built to production environments and serve it to be accessible from software clients and applications.3.Model monitoring must be conducted to automatically detect model degradation and performance issues. As such, when anomaly behaviours are identified, models can be re-trained.4.Model governance allows the tracking of models. The main approach is the provision of a model catalogue where related metainformation could be associated. Thus, models can be easily identified and found. In addition, having a suitable model catalogue paves the way for establishing authentication and authorisation policies over the models, as well as supporting auditing mechanisms.

Figure 1 shows a proposal of the ML life cycle where these challenges are considered. Due to the importance of this life cycle, this paper is focused on the packaging, deployment and serving phases.

In addition to the necessity of putting the model into production, frequently, once models are built, additional testing is required to guarantee integration with other systems. Moreover, the strategy can involve not only the production of a model, but also the execution of several chained steps involving acquisition, processing or data preparation as shown in Figure 1—that is to say an analytic pipeline. For this purpose, it can be especially useful to test such steps isolated in different hosts to better represent the target environment and to be able to focus on each step of the process separately in the evaluation phase. For these tasks, the creation of a development environment, simulating the production one, can be useful to evaluate the whole pipeline to better understand how the steps are behaving in an isolated environment.

On the other hand, the emerging edge and fog computing paradigms combined with cloud computing and on-premise servers provide some benefits. For instance, instead of executing the whole pipeline in the cloud, some time-critical steps could be executed in the edge (where data are generated) to reduce latency. Moreover, several aggregations could be performed in the fog since it normally receives data from the edge devices of the same geographical area. However, this approach also implies additional challenges. One of them involves the adequate allocation of the available edge, fog, and cloud storage and computing resources in an optimised fashion, to maximise the throughput and performance of the analytic pipelines and, at the same time, minimise the latency and budget. In addition, the availability of a heterogeneous infrastructure becomes even more complicated to manage since more diverse and advanced skills are required.

Another challenge is that in some situations the required infrastructure might not previously exist. This creation process is not trivial, and an experienced software engineer is required to perform this task. Moreover, the automation of this process is highly recommended to easily replicate environments, enhance their maintenance, and avoid human errors. In addition, independently, whether it is available or not, specific applications and configurations must be carried out (ideally, also in an automated fashion) to guarantee the adequate execution of the analytic pipelines. Consequently, the effort required to achieve this objective is bigger.

Summarising, the process to deploy ML models or analytic pipelines into both testing and production environments significantly depends upon the data engineers’ expertise. Furthermore, before being able to start the deployment stage, target machines must be created and carefully configured. Consequently, the use of automated processes and usable software clients to facilitate these tasks to non-expert users would be of enormous help. Unfortunately, this is not generally the case.

### 1.2. Contribution

This paper addresses some of the previously stated challenges. Concretely, it is focused on packaging, deploying, and serving phases of the ML life cycle in heterogeneous environments where edge, fog, cloud, and on-premise computing layers are involved, as well as in the automatic generation and configuration of missing infrastructure to deploy ML models or analytic pipelines. Moreover, the testing of analytic pipelines is addressed.

The main contribution of this research is Pangea, a tool which is able to automatically generate execution environments and deploy ML models or analytic pipelines on them for both development and production. In development, the generated environment is suitable for testing and benchmarking the behaviour of the ML models and analytic pipelines; in production, however, Pangea can create from scratch a heterogeneous infrastructure, adapted to the specific needs of each pipeline step, considering edge, fog and cloud computing layers, as well as on-premises infrastructures. For this purpose, Pangea follows these distinct steps: (1) Creating the required infrastructure if it does not exist; (2) Installing and configuring the necessary software, dependencies and underlying libraries; and (3) Deploying and serving the ML models or analytic pipelines. As such, the knowledge of data engineers is packaged as a software tool minimising the interactions required with data engineers. Thus, the distinct DevOps and MLOps techniques required to address these challenges are unified to avoid data scientists to deal with these complexities.

### 1.3. Motivation Example

This section shall put the proposed approach into an industrially relevant context, demonstrating how Pangea can be used in the mining industry, specifically underground mining. However, it is worth noting that the use of Pangea is not restricted to this specific domain. Concretely, any domain requiring the deployment of machine learning models or analytic pipelines into production environments could benefit from the features offered by Pangea. For instance, in [7], PADL (the underlying Pangea language for defining analytic pipelines) is proposed to deal with two scenarios associated with the food control and the waste management areas. In addition, this scientific paper analyses a set of projects in the smart cities domain in which the benefits of the deployment of analytic pipelines are promoted.

The mining industry is facing many challenges in terms of productivity and safety. Mining for the raw materials necessary for satisfying our daily needs is requiring that we mine at ever-greater depths. This is causing increasingly complicated geotechnical conditions, i.e., increased risks of rock bursts and other seismic events [8]. Highly sophisticated mining designs in combination with rock support methods are commonly applied to overcome the described challenges. Amongst others, these methods comprise shotcrete, wire mesh, and rock bolts. They are all designed to maintain the shape and structure of underground buildings. The spacing of rock bolts may be as little as on a 1 m × 1 m grid, leading to a total sum of tens of thousands of rock bolts being installed annually in any given underground mine. Depending on the mining system and general geotechnical environment, the support measures experience high stresses, eventually leading to the deformation, damage or even destruction of the installed rock bolts [9,10]. The status of a single rock bolt, or a group of rock bolts installed in underground mining operations, may hence provide essential information on the status of this mining section [11]. Information may contain integrity as well as stresses and forces acting on the bolt at a given position and time in reaction to the progression of the mining operation in space and time. With decreasing prices of small, versatile and reliable sensors (IoT devices) [12], the measurement and observation capacities for these rock bolts are increasing significantly.

This generates so-called “intelligent” or “smart” rock bolt [13,14] as a cornerstone for generating smart mining environments, reducing risks and increasing productivity. The information of these sensors may be complemented by additional data from moving equipment (haul trucks, drill rigs, loaders), as well as other sensors measuring environmental parameters that generate a heterogeneous set of available information. Combining this vast amount of different data sources and feeding them back to the mine management system ultimately provides the mine management, technicians and supervisors with an easy-to-use, fast and reliable decision-making tool.

For this reason, edge devices are installed along the mines to manage the data coming from the intelligent rock bolts system, as well as additional sensors installed in various other positions and equipment. Moreover, for each mine, a computer acting as a fog infrastructure is provided to control the data coming from the edge devices. Finally, managers have created an account in a cloud provider to deploy heavier workloads aiming at analysing the data.

In order to fulfil their requirements, they decided to use Pangea to deploy an analytic pipeline in its existing software infrastructure (edge devices and fog computers) and in the missing required cloud infrastructure that Pangea will automatically create in the selected cloud provider. Following this objective, a data scientist belonging to the mining company defines the pipeline below, which is illustrated in Figure 2:1.Acquisition phase: data from intelligent rock bolts are collected.2.Data cleaning phase: non-necessary fields are discarded to minimise the amount of data to transfer.3.Data cleaning phase: both null values and bad readings, not covering a defined pattern, are filtered.4.Data analysis phase: a previously trained classification model is executed to identify the non-safe areas in the mines.

Once the pipeline is defined, the data scientist has two options to provide the necessary model to deploy: create a portable format for analytics (PFA) document [15] with the model or upload the code to be deployed to a Git repository (a technology aiming at tracking and storing software changes into a centralised system, thus boosting the team collaboration). Then, using a web client created for Pangea, the data scientists creates and submits to Pangea, in development mode, a PADL document where the whole mine-related pipeline is modelled. After this process, a docker-compose document is generated, simulating an environment where each step of the pipeline and each required message queues are deployed in a dedicated container. In this development mode, the adequate behaviour of the pipeline can be tested and optimised.

Subsequently, the data scientist schedules a brief meeting with a data engineer to create the infrastructure and the deployment map (a document where the pipeline is matched with the infrastructure) documents by again using the web client. The infrastructure document will contain the characteristics of each available node and the deployment map document indicates where to deploy each step of the pipeline. Each map can be mapping (an existing node is used to deploy that step of the pipeline) or *prescription* (new infrastructure will be automatically created to deploy such a step of the pipeline). Then, the three documents, alongside the cloud provider credentials, are submitted to Pangea in production mode. In this moment, the Pangea process starts and creates the necessary nodes whilst considering the steps of the pipeline identified in the deploy map document as *prescription*. Afterwards, all the nodes involved in the pipeline are configured to be managed using Ansible. Pangea then installs the required dependencies in the corresponding nodes and copies or downloads in each node the source code to be executed (either a base code with a PFA engine encapsulated or another code from a Git-compliant repository). Finally, the code is executed in each node with the specific execution parameters defined in the PADL document.

This way, the whole pipeline is deployed in the infrastructure already available or created on the fly. Consequently, the data engineer role dependency is drastically reduced to put analytic pipelines into production environments. In Section 5, this example will be detailed from an implementation perspective to better demonstrate the potential of Pangea.

### 1.4. Structure

The rest of the paper is organised as follows. In Section 2 and Section 3 the background and related works are discussed, respectively. Subsequently, Section 4 is focused on explaining the tool: firstly, an overview including the allowed processes and flows is provided (Section 4.1). Then, the compatible models (Section 4.2) and the architecture (Section 4.3) are described. Afterwards, modifications taken into consideration to extend the underlying analytic pipeline description language are proposed (Section 4.4), and finally, the proposed web client (Section 4.5) and implementation aspects are examined (Section 4.6). Moreover, Section 5 proposes a complete scenario to validate the tool and shows some performance metrics. In Section 6, conclusions are provided and finally, Section 7 provides an outlook and perspective for future work and improvements.

## 2. Background

This work is related to two distinct research fields: (1) the automatic generation of new and existing infrastructure, since no available nodes must be created to successfully deploy analytic pipelines. In addition to the automatic preparation of necessary software and libraries and its required configurations; (2) the MLOps phase of puts analytic pipelines into production since the main objective of this approach is the deployment of analytic pipelines in heterogeneous infrastructures. Therefore, this section examines existing works from both areas.

### 2.1. Infrastructure Automation

This subsection examines the software tools belonging to the DevOps paradigm. Particularly, there are those dedicated to programmatically creating environments and those in charge of—also programmatically—installing and configuring software in remote machines. This emerging trend of creating, provisioning and configuring heterogeneous environments by using code blocks is denominated as infrastructure as code (IaC) [16].

The proliferation of diverse cloud providers and the increase in data volumes necessitates the existence of software tools able to configure a vast number of nodes with different or similar configuration in an automated way. Having this goal in mind, several tools have been created, among them, Chef [17], Puppet [18], Salt [19] and Ansible [20]. All these tools support high availability and scalability. When the master goes down, diverse mechanisms to supply it are considered, in turn, it is quite straightforward to move from a fifty nodes scenario to a five hundred nodes one. Regarding the installation, all of them except Ansible require a software agent to be installed in the nodes to be managed, by contrast, Ansible relies on SSH (Secure Socket Shell) login without requiring the installation of additional software in the administrated machines. In Chef and Puppet, the configurations are requested from the nodes to the master in pull mode and in Ansible and Salt, the master pushes the configurations to the managed nodes in an easier manner. On the other hand, Ansible and Salt are based on Ai not Markup Language (YAML)) which is quite easy to learn, but Chef uses Ruby Domain Specific Language (DSL)) and Puppet uses Puppet DSL which are more complicated. In Table 1, a comparison considering the most relevant features of these technologies is provided. Despite all of them being capable of satisfying its requirements, Pangea utilises Ansible as its underlying provision mechanism. This is due to the fact that Pangea is a tool which is capable of centralising the orchestration of a given pipeline which is aligned with the push mode followed by Ansible. In addition, before being able to use any of these tools, Pangea requires that some programmatical configurations in the nodes are made. Ansible only requires an SSH configuration which can be made by using mature Java libraries. Finally, YAML configurations minimise the learning curve.

Terraform [22] makes use of declarative configuration files to manage the interaction with cloud APIs in order to control the workflows of diverse cloud services by offering the main cloud technologies for providers to interact with. Thus, the processes of creating, destroying and updating cloud machines can be accelerated by minimising the dependency of specific providers. Therefore, Pangea makes use of Terraform to create the missing infrastructure because of the big number of connectors available to reduce vendor locking. Other alternatives for managing infrastructures are CloudFormation [23] and OpenStack Heat [24]. However, these technologies were discarded because the former is only compatible with Amazon Web Services (AWS) and the latter is only compatible with OpenStack.

Additionally, it is worth mentioning complementary technologies such as big data platforms which can execute heavy, fast and scalable processes by deploying a specific technological stack into a pre-existing and previously configured set of nodes. For instance, Cloudera Distribution Hadoop (CDH) [25] is a multi-environment analytic platform based on open source technologies. It offers an enterprise data cloud to execute scalable and elastic workloads. In addition, it offers Edge and AI support. 1010data [26] unifies data and analytics on its platform, allowing users to perform analysis on data in the same place as it is stored. Pivotal big data suite, a solution for agile data, can be deployed as part of pivotal cloud foundry and platform as service (PaaS) technologies, on-premise and in public clouds, in virtualised environments, on commodity hardware or delivered as an appliance. Azure HD insight [27] is a PaaS solution offered in the Azure cloud to execute open source big data technologies. These technologies have provided a source of inspiration in the area of deploying specific services in already created environments. In this line, Pangea could be integrated with them to provide an added value by integrating a technology in their stack that is able to deploy analytic pipelines.

Another set of interesting related technologies is that of workload orchestrators. In this field, Verma et al. [28] described a cluster manager for running large workloads on thousands of machines. This project was the basis of the popular Kubernetes technology [29]. Similarly, in the area of workload orchestration in distributed environments, Docker Swarm [30] must be considered. It offers fewer functions than Kubernetes, but with a smaller technological footprint and with the advantage of being easier to use. However, neither specifically focus on AI nor on edge device management. Other tools such as KubeEdge [31] try to bring orchestration closer to devices on the edge but again are not specifically focused on deploying analytic pipelines. Container-based technologies provide a set of advantages when deploying models or software artefacts such as the portability or the encapsulation of required libraries and configurations. Consequently, future versions of Pangea will allow for the deployment of models and analytic pipelines into containers. However, the use of containers will not be mandatory as it is in other MLOps technologies, since specific edge hardware architectures do not adequately support them. Finally, it is also worth mentioning approaches such as Apache Airflow [32] that programmatically manage the workflow life cycle.

### 2.2. Analytic Pipelines

In the field of deploying ML models and pipelines, significant works can be highlighted. There have been some efforts to create model interchange formats that are independent of tools, applications and systems. For instance, portable format for analytics (PFA) [15] is a language for defining analytic models. With PFA, when a model is produced, developers can deploy a model by using a Java or Python provided library that is able to process the model definition and execute the corresponding scoring engine. Prior to PFA, the XML-based language predictive model markup language (PMML) [33] was conceived. PMML is also independent of specific technologies to define predictive and descriptive models. As stated in [15], PFA enhances PMML capabilities to allow an extensible language to integrate new models without having to update the base scoring engine. In addition, pre- and post-processing features are supported by enabling model composition and chaining. Finally, PFA is compatible with modern Big Data technologies. On the other hand, PADL [7] enables defining analytic pipelines taking infrastructure aspects into consideration. Each step of a pipeline is associated with a model to be deployed which can be a PFA model. In this work, PADL was selected as analytic pipeline description language and, in turn, PFA is also supported, as it has an integrated PFA engine. Open neural network exchange (ONNX) [34] is another open format to represent machine and deep learning models which supports several AI frameworks. The support for this format will also be considered in future versions of PADL.

## 3. Related Works

In this section, several technologies belonging to the same research areas of Pangea are evaluated. In addition, Table 2 shows a summary of the key features taken into consideration for this analysis. Concretely, the categories considered for the comparison are:Generate infrastructure: the ability to create the necessary but not previously available infrastructure in cloud providers.Provision machines: being able to remotely configure, prepare and install libraries and software.Deployment of analytic pipelines in distinct machines: some approaches can deploy a model in a machine or deploy the whole analytic pipeline in the same machine. In this category, the objective seeks to be able to deploy each step of a pipeline in a separate machine.Edge, fog, cloud support: the possibility to somehow deal with these three computing layers.

MLFlow [35] is an open source platform that helps in the ML life cycle by addressing challenges such as experimentation, reproducibility and deployment. In addition to tracking and packaging functionalities, it enables the deployment of MLFlow models served by means of a REST API. Based on MLFlow [35], Scanflow [36] is an MLOps platform to deploy and train models on top of Kubernetes. However, identically to MLFlow, only deployment in REST API is supported. Consequently, the deployment of analytic pipelines is not considered. Clipper [37] is a prediction system that can leverage the different machine learning frameworks for model development and packaging, and it provides means for the communication of such models and the applications through a REST API. It does not support packaging in streaming mode. ML.NET [38] is an open source framework proposed by Microsoft to integrate models in applications and build pipelines. In [39], the authors presented three distinct approaches for serving models in production environments by using ML.NET as an underlying technology. In turn, this allows the authoring and deployment of models built using the .NET stack. Zoo system [40] is devoted to deploying data analytic services, mainly in edge devices, whilst considering their intrinsic constraints. For this reason, it provides a type of safety domain-specific language. The Zoo workflow is divided into two steps: development and deployment. In terms of deployment, Zoo supports Docker, JavaScript and MigrateOS. As far as we are concerned, unlike Pangea, these systems seem to not be able to deploy each step of an analytic pipeline in different machines, but the whole pipeline in a unique one. In addition, Pangea also supports the creation of missing infrastructure. PyCaret [41] is a lightweight open source Python library with the purpose of preparing and deploying models. However, it is limited to notebook-based deployments. Seldon [42] is an MLOps solution that provides the deployment of ML models in Kubernetes and ML governance capabilities. It also supports the concept of workflows or analytic pipelines but again, as far as we are concerned, the pipeline cannot be deployed in different machines.

Edge cloud orchestrator (ECO) [43] is an architecture for ML deployments in edge and cloud environments that considers different ML engines such as Spark, Flink or TensorFlow. ECO defines intelligent overlay networks (IONs) containing directed acyclic graphs where each task can be executed in different nodes. In contrast, Pangea can not only execute a step of a pipeline in distinct nodes, but it can also automatically create the required infrastructure when necessary. This way, data scientists do not have to deal with the creation of the missing required nodes.

Stratum [44] is an event-driven big data-as-a-service for Internet of Things (IoT) analytic life cycle management platform. It follows the model-driven approach of provide a declarative way to specify application and infrastructure requirements. A graphical interface is provided to compose and deploy ML models in an abstract way and then the final code is generated. Stratum can utilise a set of predefined ML algorithms encapsulated in Linux containers to compose the pipelines. In contrast, in our approach, in addition to not being limited to a subset of ML algorithms, references to Git code are supported to deploy code blocks where ML algorithms can be provided. In addition, PFA-based models can also be directly referenced since a specific code project was prepared with an encapsulated PFA engine.

On the other hand, several model deployment solutions are constrained to the specific ML library used during the training phase. For instance, Torchserve [45] and Tensorflow Extended [46] belong to Pytorch [47] and Tensorflow [48], respectively. Consequently, in addition to not being conceptualised to manage the underlying infrastructure, these technologies cannot be used as a general purpose tool to deploy models.

Another interesting work is that of Kubeflow [49], which was focused on the deployment of analytic pipelines/workflows in Kubernetes clusters. This is a promising technology, but it currently does not provide support for restricted layers that do not have a Kubernetes cluster, such as the edge and fog computing layers. Conversely, Pangea can deal with operating systems without having a workload manager installed indistinctly of the computing layer.

Among the studied technologies, none cover more than two categories apart from STRATUM. However, as previously commented, it is limited to a subset of ML algorithms. Consequently, it can be concluded that there is an existing gap to create Pangea as a unified tool capable of managing the required infrastructure, provisioning it, and deploying analytic pipelines in different nodes belonging to the edge, fog and cloud computing layers.

## 4. Pangea in Deep

### 4.1. Overview

This paper describes the conceptualisation of a software tool, denominated Pangea, that aims to automatically deploy analytic pipelines in distinct environments (edge, fog, cloud and on-premise). The motivation for creating Pangea lies in facilitating the operationalisation of ML models and pipelines for non-expert users. Therefore, from its conceptualisation, specific building blocks (see Figure 3) for addressing key MLOps features were sketched out. In Figure 3, these preliminary building blocks are shown.

For this work, PADL syntax [7] was embraced. PADL enables the description of the different steps composing an analytic pipeline and the communication mechanisms among them. In addition, among other features, it supports establishing the model to be deployed and the input–output parameters required to execute it. PADL is technology and infrastructure agnostic, which makes it suitable for considering the desired environments (edge, fog, cloud and on-premise). Therefore, each step of the pipeline can be separately deployed in any node of a given heterogeneous infrastructure.

Figure 4 presents the development and production modes defined for assisting users in the deployment of analytic pipelines.

The development mode aims to generate a container infrastructure wherein the diverse steps of a pipeline and the necessary communication technologies are deployed. As such, the analytic pipeline can be prepared in a simulated local environment for testing and benchmarking purposes. The input consists of a PADL document describing the steps of the pipeline to deploy. Then, a process is triggered to create a docker-compose file [50] that data scientists could execute to have the pipeline deployed for local testing. The automatically generated docker-compose file is composed of a service for each step of the pipeline and services for simulating the communication mechanisms among the steps. Figure 5 shows this process.

The production mode creates the necessary non-existing infrastructure, provisions and configures the required software and libraries in the target nodes and finally, deploys and serves the pipeline into production. In addition to a PADL document, two additional documents must be provided to accomplish this process (see Figure 6):Infrastructure document describes the infrastructure by listing the available nodes and their required properties such as universally unique identifiers (UUID), hostname, resources (CPU, number of cores, disk and memory) or root credentials for SSH access.Deploy map document: defines a mapping between the PADL and the infrastructure documents. Specifically, the hosts where the steps of the pipeline and the communication technologies must be deployed in. If the steps of the pipeline do not have any host assigned due to insufficient pre-existing infrastructure, a suitable infrastructure will be automatically generated for such steps.

Examples of the use of these three inputs are provided in Section 5.

### 4.2. Compatible Models

Data transformations to be executed in each step of the pipeline can be provided in diverse ways. The first objective was to be compliant with PFA scoring engines since it is a technology-independent format and the framework used in the training phase. For this purpose, a base code with a PFA engine was built. This code supports the message queuing telemetry protocol (MQTT) and Kafka producers and consumers in order to manage data, as well as the capacity of reading log files. As such, when executing this code, in addition to the PFA document to execute, the types of input and output technologies are passed as parameters and the adequate consumer and producer will be utilised. This code has two versions: one for edge and fog environments and the other one for the cloud, which integrates real-time processing-distributed technology. In addition, for both versions, a docker image was prepared for creating the necessary containers in the development mode.

On the other hand, the code available in Git repositories is also supported. This code should provide the necessary consumers and producers, as well as a dockerfile, to be compatible with the development mode.

### 4.3. Architecture

Figure 7 illustrates the proposed architecture derived from the building blocks sketched out in Figure 3. First, a web client was built (described in Section 4.5) to enable users a graphical interaction and a command line interface (CLI) client is planned to be developed.

Then, the Orchestrator receives the queries, parses and analyses them and identifies which internal processes should be triggered in order to be able to decide:1.Whether the analytic pipeline requires being deployed in a development or production environment.2.Whether it is necessary to create additional infrastructure and with which technologies or, in contrast, there is already sufficient infrastructure.3.The configurations, dependencies and code required for executing each step of a given pipeline.

The tasks coordinated by the Orchestrator are conducted in diverse components. The Infrastructure Generator, if necessary, creates specific machines to deploy the pipeline steps and the communication technologies by using the infrastructure as code (IaC) paradigm. The Provisioner installs the required software in each host considering aspects such as the target operative system, the software version to install or the available host resources. In addition, it deals with the specific configurations that should be made both in the host operating system and its applications. Once the environment is prepared, the Pipeline Deployer deploys each step of the pipeline in the corresponding host. A step of the pipeline may be considered as a distributed task and consequently, such a step would be deployed in various coordinated hosts.

Previously explained components make use of the Connectors Manager to establish communication with the hosts. This component must support connectors for most market-used technologies in the field. By using a decoupled component to manage the connectors, future technologies easily could be integrated. It is worth mentioning the SSH support since the provisioning and configuration steps will be made using this protocol.

Finally, different mechanisms for persisting information must be supported to enable the flow of data among the pipeline steps. Therefore, the component Data Source Manager is responsible for providing access to retrieving and storing information from the most used sources of information technologies. Concretely, a wide range of persisting technologies must be compatible to deal with the main architectures of the market:RDBMS (Relational Database Management System) includes traditional databases for managing static data.NoSQL for scalable computations and support of distributed data.Time Series to enhance the management of temporal data.Search Engines for supporting complex data searches.Publish/subscribe technologies since they are widely used in real-time applications.File processing must be supported to deal with local or remote logs, comma-separated values (CSV), JavaScript Object Notation (JSON) or extensible markup language (XML).

### 4.4. Augmenting PADL with Expression Language

As stated in Section 4.1, a PADL document is the main input of the system. PADL allows one to define an entrypoint with a command and parameters to run a specific code. The problem raised was that some of the values of the parameters could not be set when defining the PADL document. For instance, the folder where the executable will be stored is not necessarily known by the data scientist nor the hostname of the created infrastructure. Consequently, a preliminary expression language was defined to allow setting expressions instead of fixed values which will be evaluated by Pangea during the process. It is worth highlighting that, in development mode, random values are generated to supply such parameters since there is no infrastructure involved in such a phase.

Concretely, the expression VAR{PROJECT_FOLDER} targets the folder where the code to execute is downloaded. Each expression must have the following form {$EXPRESSION}. In addition, these expressions can reference each of the three main inputs (PADL, infra and deploy map) using the following syntax doc(TYPE_OF_DOCUMENT). It is also allowed to reference the current element in a list by using the reserved word self. Finally, expressions can be concatenated with the + symbol. In Figure 17, the following excerpt of PADL provides an example using the expression language for setting the values of an entrypoint.

**Listing 1 sensors-22-04425-f017:**
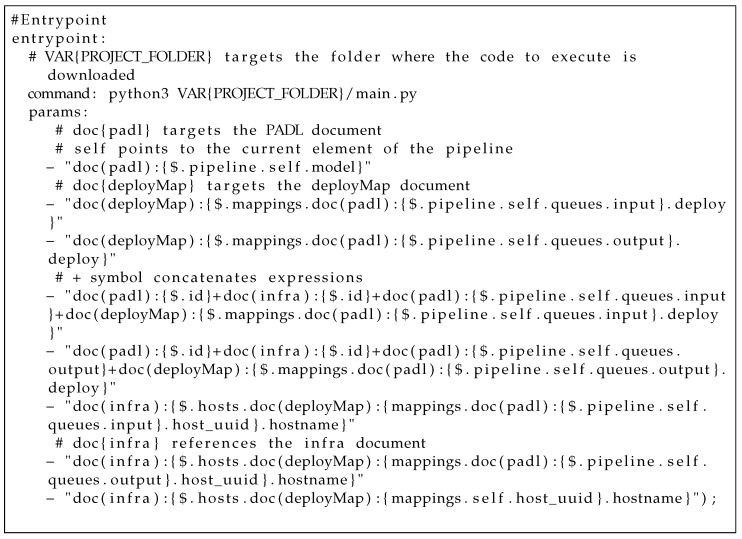
Example of PADL expression language.

### 4.5. Web Client

A web client was created to facilitate end-users’ interaction with the REST service. This client is an Angular-based application which integrates a JSON online editor [51] with various view modes (text, code, tree view) to provide a usable means of editing the required inputs (PADL, infrastructure and deploy map). In addition, an Angular material design stepper component was included to guide the users during the editing process [52]. Figure 8 shows a screenshot of the Angular client.

### 4.6. Implementation

For the implementation of the REST API and the different components of Pangea, Java 8 was used since it is a sufficiently mature language with sufficiently robust functionalities to create APIs. In addition, the software project was configured with an Open API 3 generator. As such, when the API is modified, the related code structures and API classes can automatically be re-generated. This allows the agile development of new functionalities.

The Infrastructure Generator module internally generates Terraform-compliant files (version 0.14.4) to generate the infrastructures required and to manage their life cycle. The Provisioner and Pipeline Deployer modules interact with a set of Ansible playbooks (version 2.9.6) dedicated to provisioning the hosts and executing the pipeline steps. The selection of both technologies is justified in Section 2.1.

The base code with the PFA engine and producers and consumers integrated (mentioned in Section 4.2) and destined for edge and fog nodes is developed in Python 3 since it is a widely used language supported by a myriad of mathematical, machine learning and statistical libraries. Moreover, the most mature PFA engine library is written in Python, which is called Titus [53]. For cloud environments, Apache Spark [54] was selected for being a unified data processing engine suitable for multiple purposes, such as batch and real-time processing, compliance with the main ML algorithms and the provision of graph-processing utilities. Concretely, the Apache Spark 3.0.1 structured streaming (in Python language, PySpark) library was selected. On-premise, depending on the available resources, the Python or the PySpark version will be deployed. Regarding the connectors, currently, both versions support the consumers and producers of log files, MQTT and Kafka. Figure 9 shows the main blocks of this code.

## 5. Validation

### 5.1. Objective and Design of the Experiment

This section provides the details for the implementation steps conducted to build the scenario proposed in Section 1.3. With this scenario in mind, we aim to validate the main contributions of this article by addressing each one of the following objectives:1.Pangea can deploy analytic pipelines in heterogeneous environments in development and production modes.2.If necessary, unavailable infrastructure must be generated.3.The interaction with Pangea proposing a Web client which uses standard communication protocols such as HTTP (HyperText Transfer Protocol) can be facilitated.

The experiment was designed to answer the following questions:1.Is Pangea able to assist in the deployment of a local pipeline for development?2.Is Pangea able to deploy a pipeline in a production environment in an edge–fog–cloud infrastructure?3.Is Pangea able to automatically create the required infrastructure?4.Does the web client provide JSON editors to build the necessary files and can the web client submit such files to Pangea?

Therefore, the motivation example described in Section 1.3 defines a pipeline composed with sufficient steps to deploy them in a heterogeneous infrastructure, both in local and in production. Moreover, alongside this information, the communication technologies will be used to define the PADL document. In addition, some virtual machines were created to simulate nodes for edge, fog, deploying an MQTT broker and deploying a Kafka broker. However, there is not any available node to simulate the cloud. Consequently, an Amazon Web Service account was created to enable Pangea to automatically create such a node as a virtual machine. Finally, during the process, the Web client must be used to interact with Pangea to test its adequacy.

### 5.2. Execution Environment

The four virtual machines for simulating the existing nodes were prepared with an Ubuntu 20.04 image using Vagrant 2.2.6. For this purpose, a Vagrantfile for each node was created, where the hostname, IP, CPUs and memory parameters are defined. The root user and password of all the machines is *vagrant*, which is used to establish the SSH connections. Figure 18 shows an example of this file and Table 3 presents the parameters of each node. It is worth mentioning that the nodes labelled as MQTT and Kafka are where the respective brokers will also be automatically deployed by Pangea.

**Listing 2. sensors-22-04425-f018:**
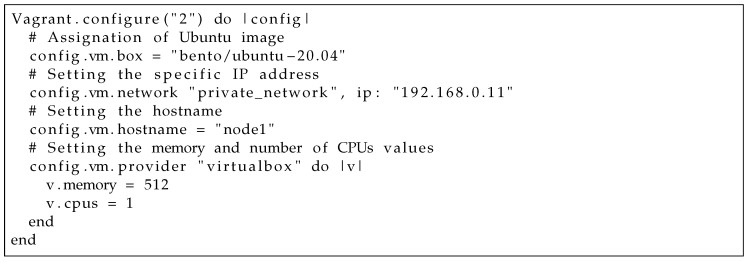
Example of the Vangrantfile used to create the necessary nodes for simulating the existing infrastructure.

Additionally, a simple shell script is provided to the edge nodes to simulate a log file where a new line in JSON format is inserted every five seconds to simulate data acquisition. Each JSON record has its form shown in Figure 19.

**Listing 3. sensors-22-04425-f019:**
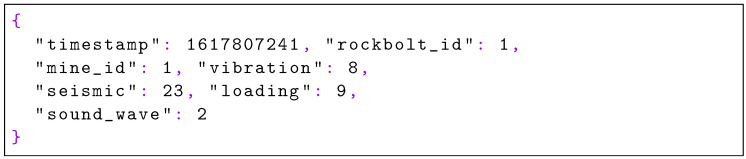
Example of the input data which includes timestamp, mine, rockbolt identification, amongst others.

The values are randomly generated, and we did not make an additional effort to provide real values because this is irrelevant to demonstrating the use of Pangea. Similarly, the thresholds and formulas defined in the PFA documents of Section 5.3 are invented, but sufficiently valid for the purpose of exemplification. Finally, the code with both versions of the PFA engine project was uploaded to an internal Git repository.

### 5.3. Execution

As defined in Section 1.3, firstly the data scientist defines the steps of the pipeline to execute in the PFA format. The preliminary task, data acquisition, is not related to a PFA document since the data are retrieved from the input queue of the first step of the pipeline. The PFA document shown in Figure 20 is in charge of discarding the unnecessary information produced by the intelligent rock bolts (IRBs) and only transferring the identification fields and the loading and vibrations measures.

**Listing 4. sensors-22-04425-f020:**
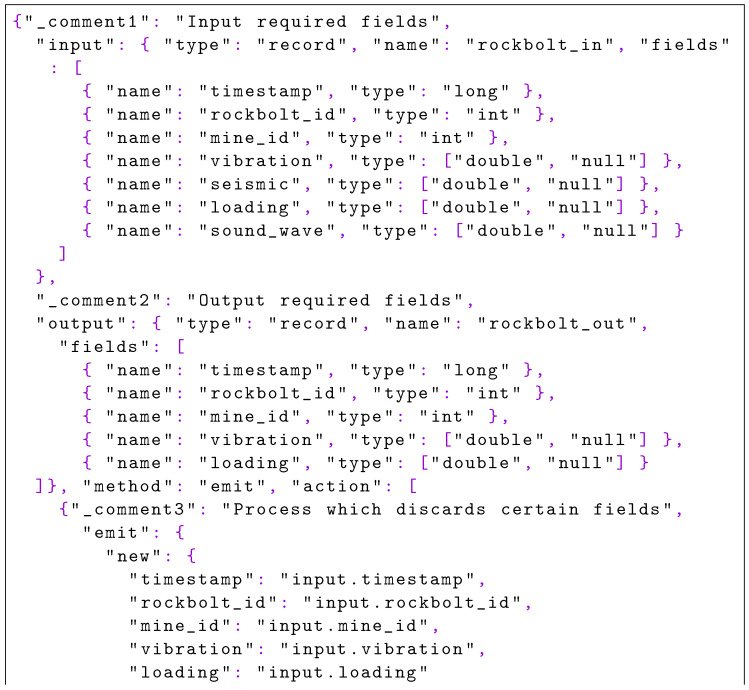
Discard fields with PFA: based on the input provided, the unnecessary fields for the processing are removed in the output.

With the code of Figure 21, records with null values either in the loading field or in the vibration field are filtered.

**Listing 5. sensors-22-04425-f021:**
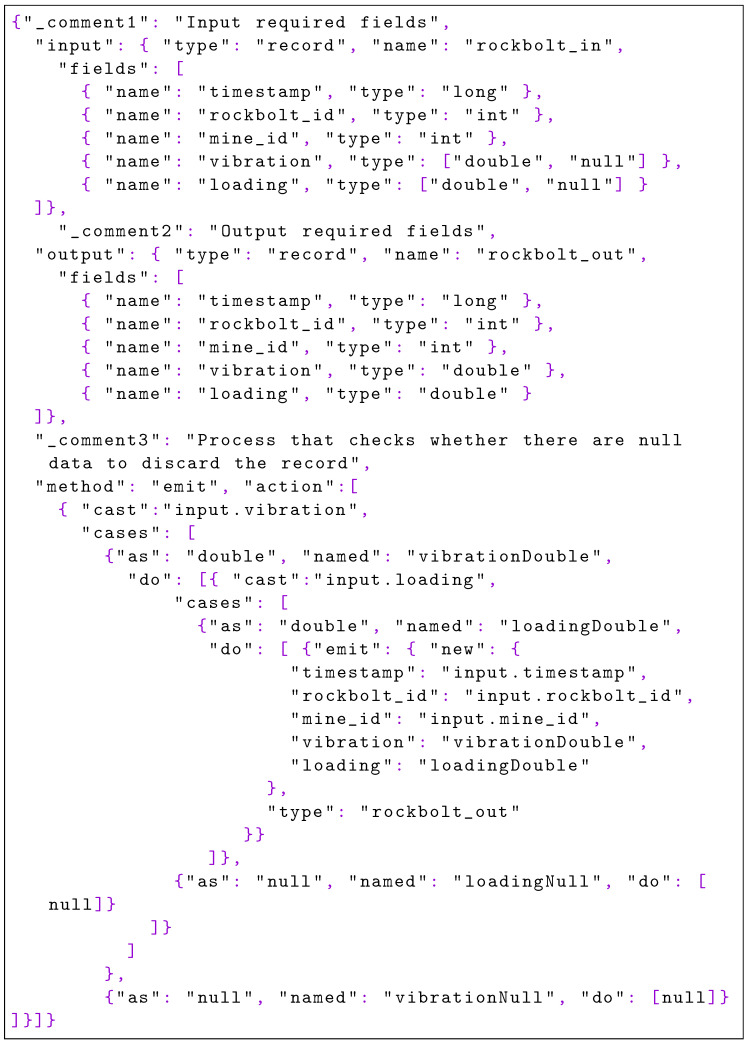
Filter nulls with PFA: this PFA excerpt accepts null values as input, but not as output; in turn, incoming null values are filtered but are not streamed to the following step of the pipeline.

Figure 22 shows the PFA document that applies specific thresholds over the fields to filter bad readings.

**Listing 6. sensors-22-04425-f022:**
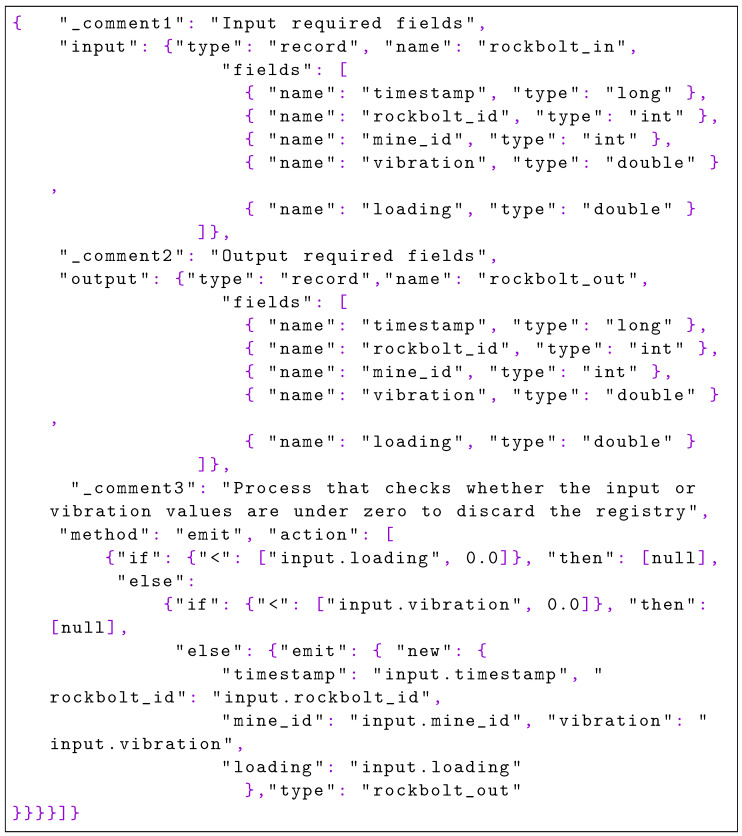
Filter by threshold with PFA: negative values are filtered as they are considered bad sensor readings.

Furthermore, Figure 23 illustrates the process followed by a random forest model to classify the zone into three states: *no risk*, *relative risk* or *high risk*.

**Listing 7. sensors-22-04425-f023:**
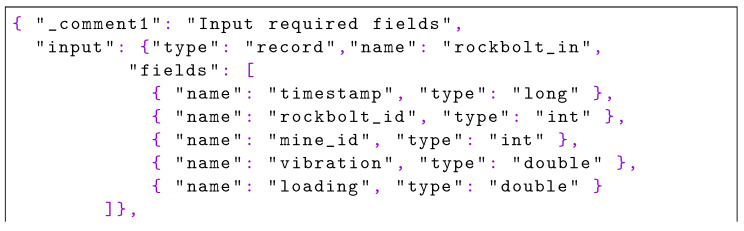
A random forest description using PFA.

Subsequently, the data scientist creates the PADL document using the Pangea client (Figure 10 shows a screenshot of the client when editing the PADL document). The *pipeline* section models the diverse steps of the pipeline, each associated with the corresponding PFA model. In addition, the input and output queues are defined for each step to identify where data come from and goes to. Finally, there is a *queues* section where the queues used in each step are defined, identifying its type and specific necessary values. For instance, in the expanded queue, *from_disk*, the path of a log file is provided which simulates the acquisition phase. Apart from the visible fields in Figure 10, there are additional fields such as the entrypoint definition as it is exemplified in Section 4.4.

Then, the PADL document is submitted to Pangea in development mode. As such, a docker-compose file is generated as shown in an excerpt of Figure 24.

**Listing 8. sensors-22-04425-f024:**
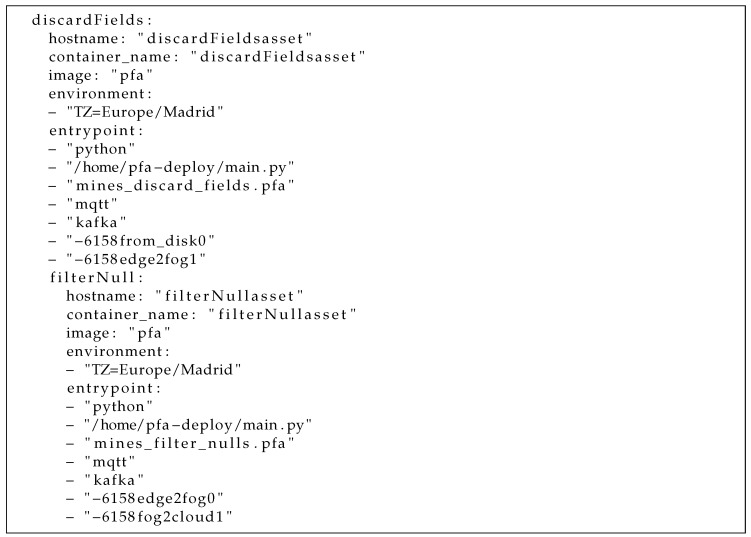
Docker-compose excerpt where a container for each step of the pipeline with its properties can be seen (note the different parameters in each entrypoint used to instantiate the container).

This file describes one container for each step of the pipeline having the parameters required for the image as an input. Moreover, there are additional containers to simulate the queues required to execute the whole pipeline. Finally, the local deployment can be achieved by executing the docker-compose file using the command docker-compose up in the data scientist’s PC. Thus, the pipeline can be locally evaluated to analyse the adequate behaviour of the models integrated together in a pipeline and the communication mechanisms. For instance, testing the whole pipeline could be beneficial to identify incompatible formats between the output of a model and the input of the following one, before deploying it into a production environment. Another advantage is to analyse whether the queue technologies are being properly used by the correspondent producers and consumers.

Once the pipeline has been verified in local mode, it is the moment to deploy it in a real environment. For this purpose, the data scientist would briefly require the help of the data engineer to create—using the Angular client—the infrastructure and deploy map documents. Firstly, the available infrastructure is described in the *infra* document, as shown in a screenshot in Figure 11. In addition to the hardware configuration, such as the number of cores and memory, the SSH configuration must be provided. It should be noted that this document was created as a matter of illustration for a research objective. For future enhancements, security will be more seriously considered. Afterwards, the deploy map document (screenshot in Figure 12) must also be built to match each step of the pipeline and each queue with existing nodes or setting them as *prescription*, meaning that they must be automatically created. In this document, the *deploy* attribute to guide Pangea where each element should be deployed is also described. Currently, the values supported for such an attribute are *edge*, *fog*, *cloud*, *disk*, *mqtt*, and *kafka*. In this way, Pangea installs the libraries and software in the correspondent node. For instance, in MQTT nodes, Mosquitto is installed, whereas in kafka nodes, Kafka and Zookeeper are installed. Another example is that in edge and fog nodes, the PFA engine base code is downloaded and utilised and, in contrast, in cloud nodes, the PFA engine is encapsulated in a PySpark Structured Streaming application where the PFA engine is integrated as a user defined function (UDF). It is worth mentioning that for this version of Pangea, Spark is deployed in a single machine as a matter of illustration. Future versions will manage the distribution of the computation between a worker and a set of slaves. As a last step, cloud-provided credentials must also be submitted to be able to generate the necessary infrastructure (note again, security will be enhanced in the future). Once the required input is built, it is submitted to Pangea using the client.

Then, Pangea starts the process and follows the steps below:Parse of the input documents.Create the required infrastructure: in this step, the infrastructure marked as *prescription* in the deploy map document needs to be generated. For this purpose, a Terraform template is provided with variables that are generated on-the-fly by examining the input documents. Then, the terraform init and terraform apply processes are triggered to create the infrastructure using such a template populated with the necessary values. Finally, the correspondent data of the recently created infrastructure are integrated in the infra and deploy maps documents.Provision: this step configures the nodes and downloads and installs the necessary code. This process can be divided into a few sub-steps:(a)If there were hosts in the infra document not used in the deploy map document, they are discarded.(b)An Ansible inventory file is generated to manage the nodes to operate with.(c)An Ansible configuration file is generated(d)SSH access is configured in the nodes to be operated with Ansible.(e)Several Ansible playbooks are executed to install the required dependencies in each node. For instance, the installation of Git, Mosquitto, Kafka or Spark.(f)The file /etc/hosts of each node is configured with the IP and hostname of the rest of the nodes involved in the pipeline to facilitate communication among them.(g)The code to execute in each node is downloaded by using the previously installed Git client. Concretely, for this example, the image of the PFA engine code is downloaded in edge and fog nodes, and the PFA engine code integrated with PySpark is downloaded in the cloud node.Deploy: firstly, the expressions defined in the parameters of the entrypoint are evaluated to extract the values and then, the code is invoked in each node by another Ansible playbook with these parameters and the command is also defined in the entrypoint.

When the process finishes, some validations are performed to verify that each step of the pipeline is deployed and that the pipeline is working correctly. Concretely, in edge and fog nodes, the command ps -aux | grep python is executed and in the cloud node, the command ps -aux | grep spark. Thus, the required processes are identified as running processes. In addition, the Mosquitto and Kafka topics are consumed to ensure that the messages are arriving using, respectively, the commands mosquitto_sub -h IP -t topic and kafka -console -consumer.sh --topic  topic --from -beginning --bootstrap -server localhost:9092.

### 5.4. Performance

This subsection provides some metrics regarding Pangea execution. Five different executions were conducted to better illustrate the time taken to deploy pipelines, the memory consumption and the CPU usage. For this exercise, no node was marked as prescription in the deploy map document (that is to say, no infrastructure will be automatically generated) to avoid additional AWS costs. The executions were performed in a machine with 64 GB of memory and 8 CPUs with 2.10 GHz where Pangea was deployed alongside the Vagrant machines described in Table 3. In addition, a node for simulating the cloud layer was initialised using another Vagrant machine with 4096 MB of RAM and four CPUs.

The outputs of the executions were redirected to a text file to easily extract the times taken. Moreover, a simple script was created to measure the percentage of total RAM consumed and the CPU usage of the current Java process every thirty seconds.

In Figure 13, the times taken by the executions are shown. The maximum value is 15.6 min, while the minimum is 14.4 min. The arithmetic average is 14.948 min and the standard deviation is 0.39.

Regarding RAM memory, the values collected every thirty seconds are a percentage of the total memory of 64 GB. First, these values were transformed to Megabytes and then the arithmetic average was also calculated. Figure 14 shows the results of this process. Executions 1, 3 and 4 consumed 0.7% (458 Mb) during the whole execution, whereas Execution 2 in the first measure consumed 0.7% and 0.8% (524 Mb) in the other ones. Execution 5 consumed 0.7% in the first four measures and 0.8% in the following ones. Consequently, the maximum value was 522 Mb, the minimum value was 458 MB, the arithmetic average was 481.8 Mb and the standard deviation was 29.287.

Finally, Figure 15 shows the arithmetic average of the CPU usage measures taken every thirty seconds. The maximum CPU usage is 7.95%, the minimum 2.18%, the arithmetic average is 5.84% and the standard deviation is 2%. However, this average of each execution is not especially representative since the CPU usage significantly varies between each execution. Therefore, Figure 16 shows a chart with the CPU measures taken every thirty seconds.

Despite not having included the infrastructure generation process in the performance tests, it is worth mentioning that this process takes between ten and fifty seconds. Most of the time is spent downloading and installing the software technologies such as Spark and Kafka and the remaining time is spent on the configuration of the machines and the pipeline deployment.

### 5.5. Summary

By proposing this step-by-step example, the diverse claims of Pangea were addressed. Concretely, each objective was fulfilled in the following way:1.Automatically deploying analytic pipelines for both contexts development and production in edge, fog and cloud infrastructures: the pipelines created by the data scientist were deployed in a heterogeneous infrastructure composed of edge, fog and cloud layers.2.Automatically generating previously unavailable infrastructure: the cloud infrastructure was not available and a virtual machine in Amazon Web Service was created to satisfy this requirement.3.Providing a usable client: the process of submitting the required documents to Pangea was conducted using an Angular client created for such an objective. As such, data scientists do not require using a command line interface to create HTTP requests or an external tool such as Postman.

## 6. Conclusions

In this paper, the Pangea tool aiming to automate the process of deploying analytic pipelines in a heterogeneous infrastructure composed of edge, fog and cloud/on-premise nodes was presented. Concretely, Pangea addresses the MLOps phases of packaging, deploying and servicing ML models and analytic pipelines in both development and production environments. In addition, the creation of missing infrastructure in cloud and on-premise environments is supported. Moreover, a Web client was developed to ease the interaction with Pangea.

In order to demonstrate the novelty of this tool, in Section 3, a substantial number of technologies were analysed and contrasted with Pangea in the text and by using a comparison table. From this section, it can be concluded that there is an existing gap to justify the innovation aspects of Pangea and its development. Concretely, the main advantage of using Pangea is the possibility of packaging and deploying analytic pipelines while creating the required infrastructure when necessary. In addition, the deployment of analytic pipelines in streaming mode, without using a REST API, and in a distributed fashion is rarely possible with the existing MLOps tools. Consequently, data analysts can benefit from using Pangea instead of a set of tools. From our perspective, the main disadvantage is not to have the possibility to monitor and retrain analytic pipelines. As such, the unification of MLOps phases into a single tool would be complete. Therefore, these considerations will be addressed in future works.

On the other hand, with the objective of showing the usefulness of Pangea, in Section 1.3, a motivation example in the mine domain is proposed where Pangea can be seen as a unified tool to encapsulate the knowledge of data engineers. As such, the interaction between data scientists and data engineers can be minimised, as well as the necessary time to put analytic pipelines in heterogeneous production environments. The use of Pangea in the mining domain can help to improve the security of mines by deploying analytics to identify the safe zones by utilising a usable and unified tool. However, Pangea is also useful in other domains requiring the deployment of analytic pipelines

Finally, Section 5 provides a step-by-step example of Pangea to validate the three main claims of this article: (1) the deployment of analytic pipelines in edge, fog and cloud infrastructures; (2) the creation of missing infrastructure in cloud and on-premise layers; and (3) the provision of a web client to ease the interaction with Pangea. Furthermore, the performance analysis carried out shows that the measures are justified from the timing perspective since there are several software artefacts to download, install and configure.

## 7. Future Work

Pangea is an ambitious tool which requires a large effort to be conceptualised and implemented. This first version is sufficiently advanced to show its potential benefits. However, we are already planning to extend it to deal with a wider range of use cases and make it compatible with more technologies and connectors.

Currently, both PADL and Pangea are not designed to support the description and deployment of analytic pipelines in the training stage. For this reason, we are already defining mechanisms to achieve this goal. As such, data scientists will be able to automatically deploy their training algorithms in real infrastructure to test them with minimum effort. Thus, Pangea will facilitate the life cycle of trendy paradigms such as deep learning, federated machine learning or transfer learning. Therefore, the ML life cycle will be better supported, including training and test phases. In addition, to complete this life cycle monitoring, re-training and model governance will also be explored in depth.

Pangea can automatically create infrastructures, but it is planned to manage the whole infrastructure life cycle including the stopping and destruction of machines. In addition, user accounts will be supported to enable users to log into Pangea and interact with their pipelines and infrastructure. As stated in the paper, cloud and SSH credentials will be treated in a more secure way.

Moreover, compatibility with new ways of providing models will be studied. For instance, in the short term, there are plans to support MLFlow models [35].

Finally, the web client will be enhanced to support the management of the users, the pipelines and the infrastructure. Additionally, a set of charts will be provided to monitor pipelines and infrastructure behaviour. Consequently, Pangea will integrate some mechanisms to advance in the monitoring field.

## Figures and Tables

**Figure 1 sensors-22-04425-f001:**
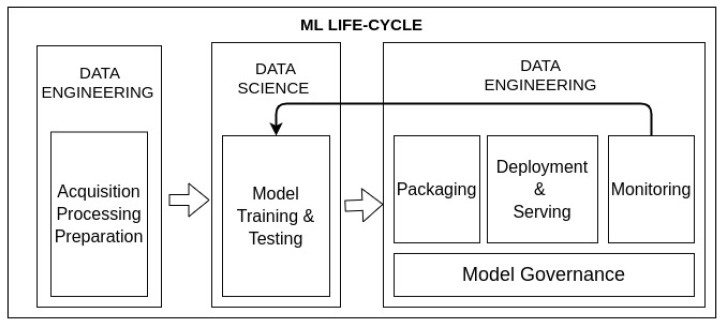
Main phases of the ML life cycle targeted at operationalising ML models in production environments.

**Figure 2 sensors-22-04425-f002:**
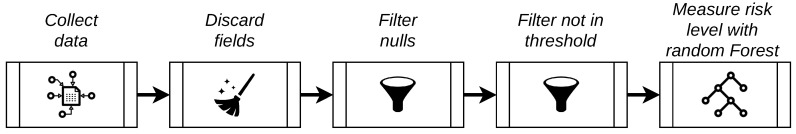
Pipeline conceptualised for the scenario in which the diverse steps proposed are illustrated.

**Figure 3 sensors-22-04425-f003:**
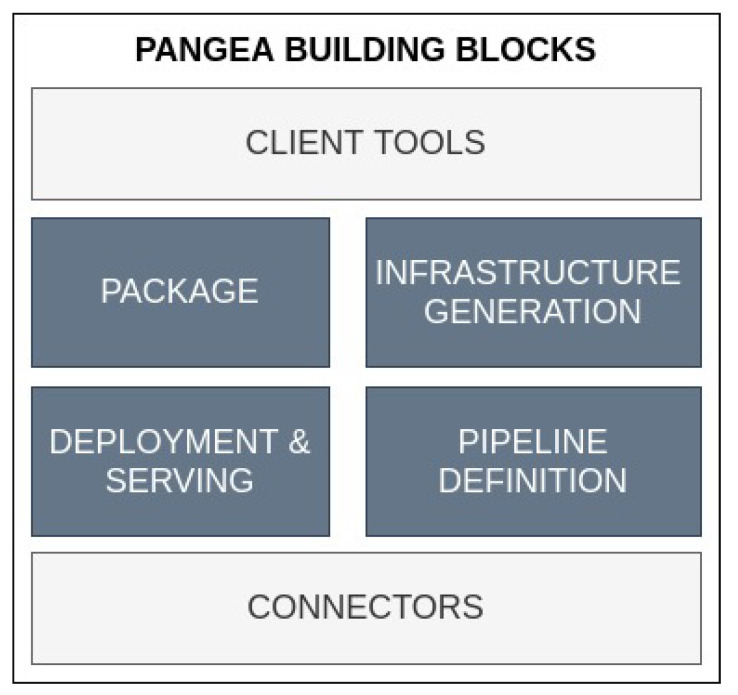
Pangea building blocks for the packaging, deploying, and serving of ML models; as well as for generating the required infrastructure.

**Figure 4 sensors-22-04425-f004:**
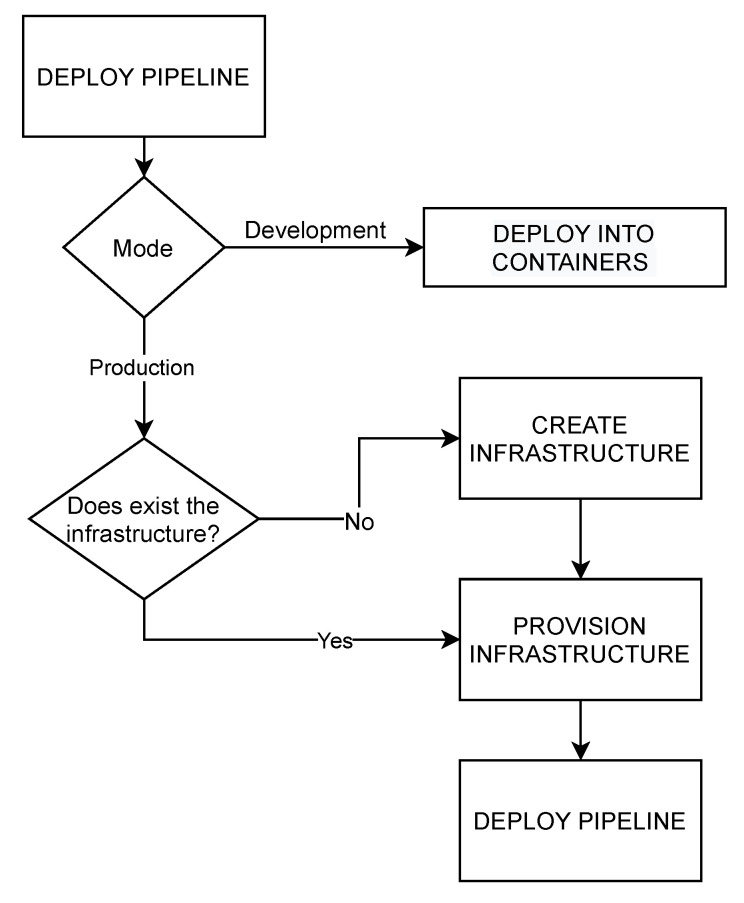
Deployment of pipelines in development or production with PANGEA.

**Figure 5 sensors-22-04425-f005:**
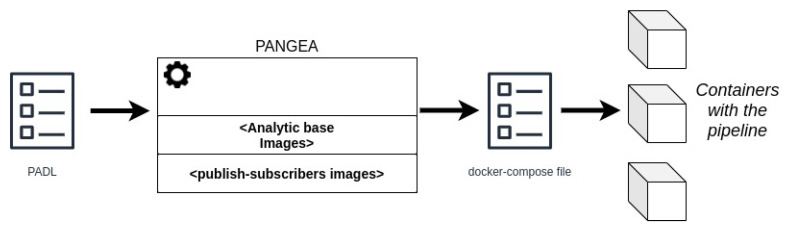
Development process aimed at automatically creating a set of docker containers, starting from a PADL document, to test and validate an analytic pipeline before being deployed in a production environment.

**Figure 6 sensors-22-04425-f006:**
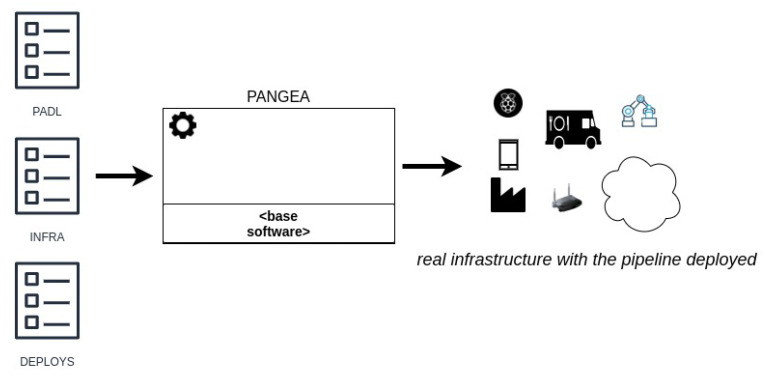
Process for automatically generating and configuring the required missing infrastructure, and packaging, deploying, and serving the analytic pipeline.

**Figure 7 sensors-22-04425-f007:**
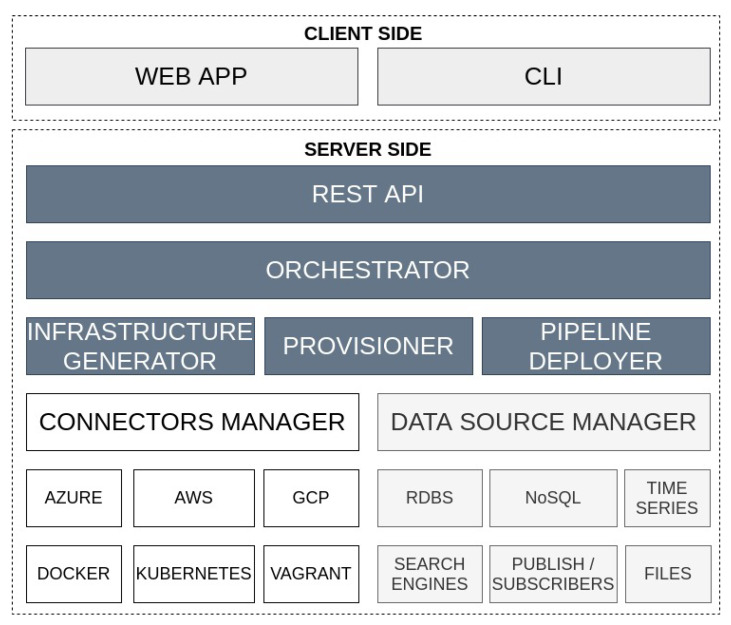
Pangea architecture overview where the specific components derived from the building blocks (see Figure 3) are materialised.

**Figure 8 sensors-22-04425-f008:**
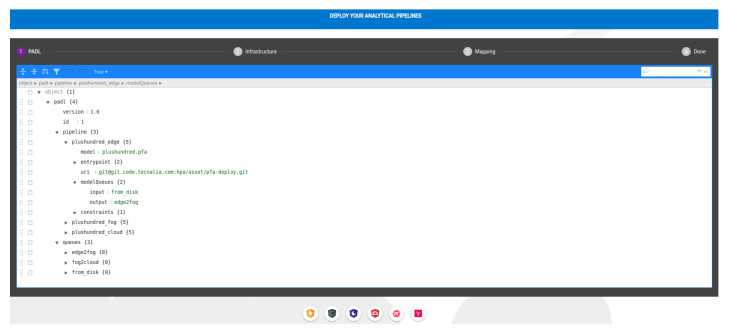
Angular client overview: this client provides a stepper component including three JSON editors (PADL JSON editor, infrastructure JSON editor and DeployMap JSON editor) and a confirmation step.

**Figure 9 sensors-22-04425-f009:**
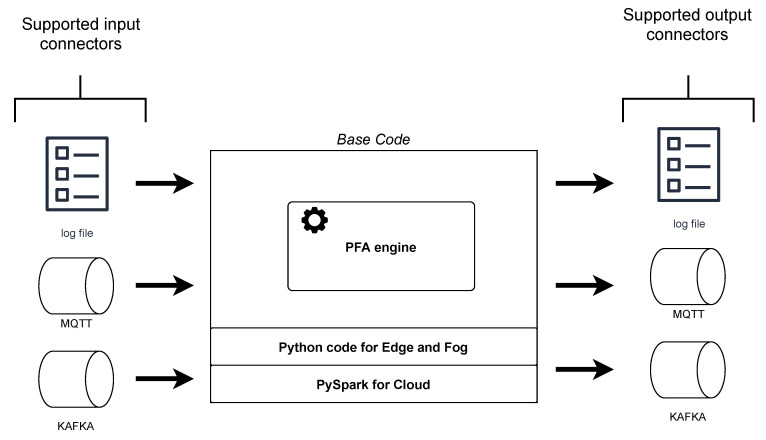
Base code created for deploying the models in streaming, alongside its different inputs and outputs.

**Figure 10 sensors-22-04425-f010:**
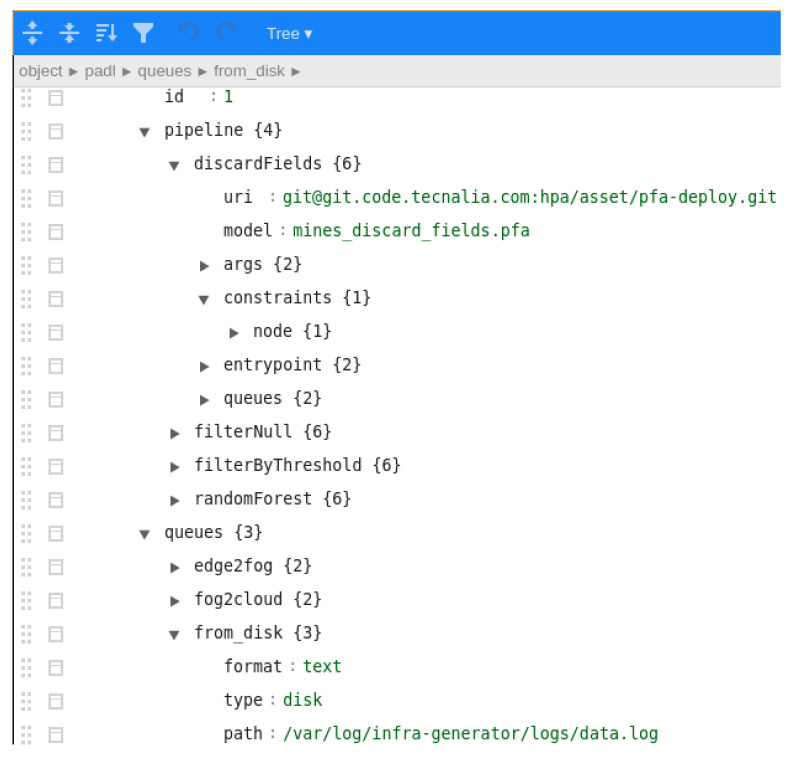
Screenshot of the client when editing the PADL pipeline definition. The four steps of the pipeline are defined and specific properties can be observed such as the URI where the code is located, the model to be executed and the queues defined.

**Figure 11 sensors-22-04425-f011:**
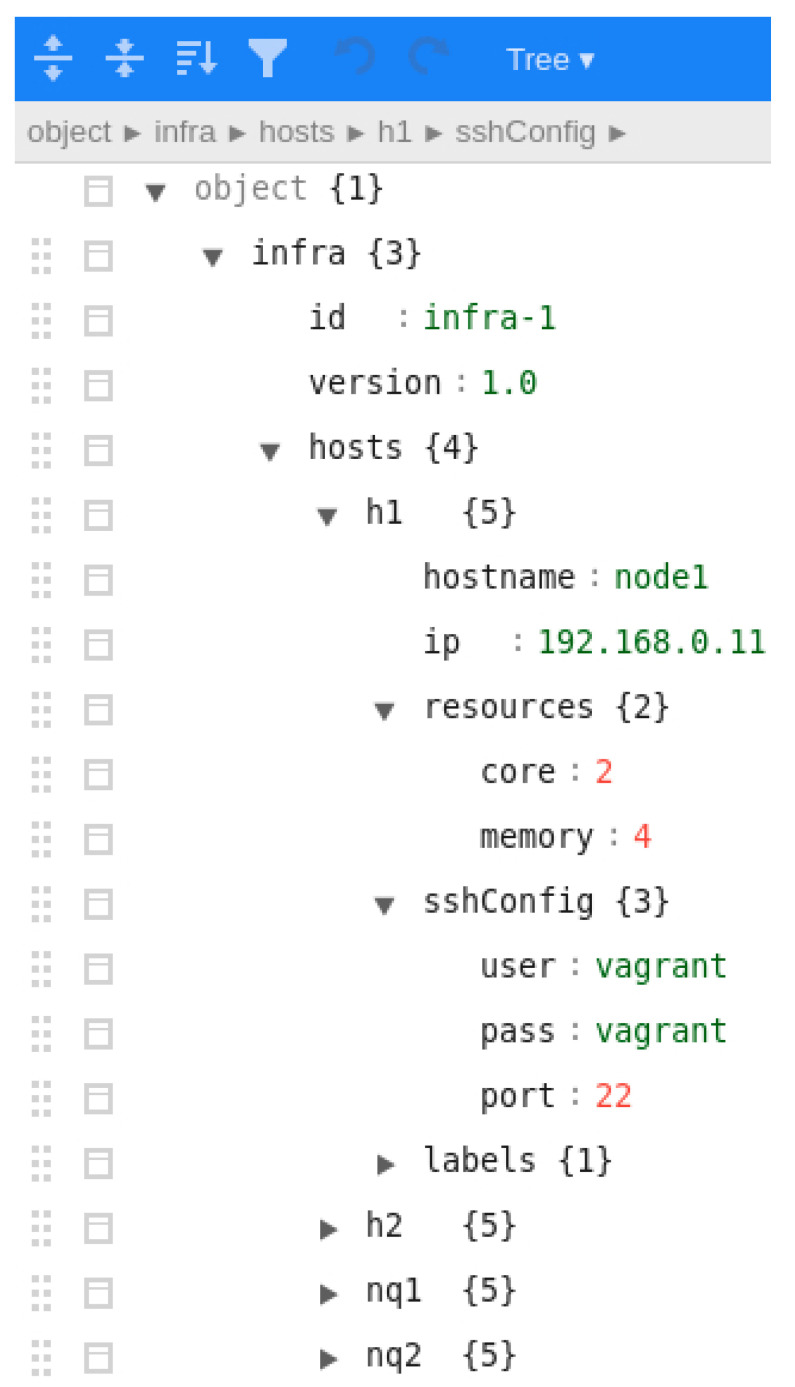
Screenshot of the client when editing the infrastructure definition. An example of the required properties of each node is represented.

**Figure 12 sensors-22-04425-f012:**
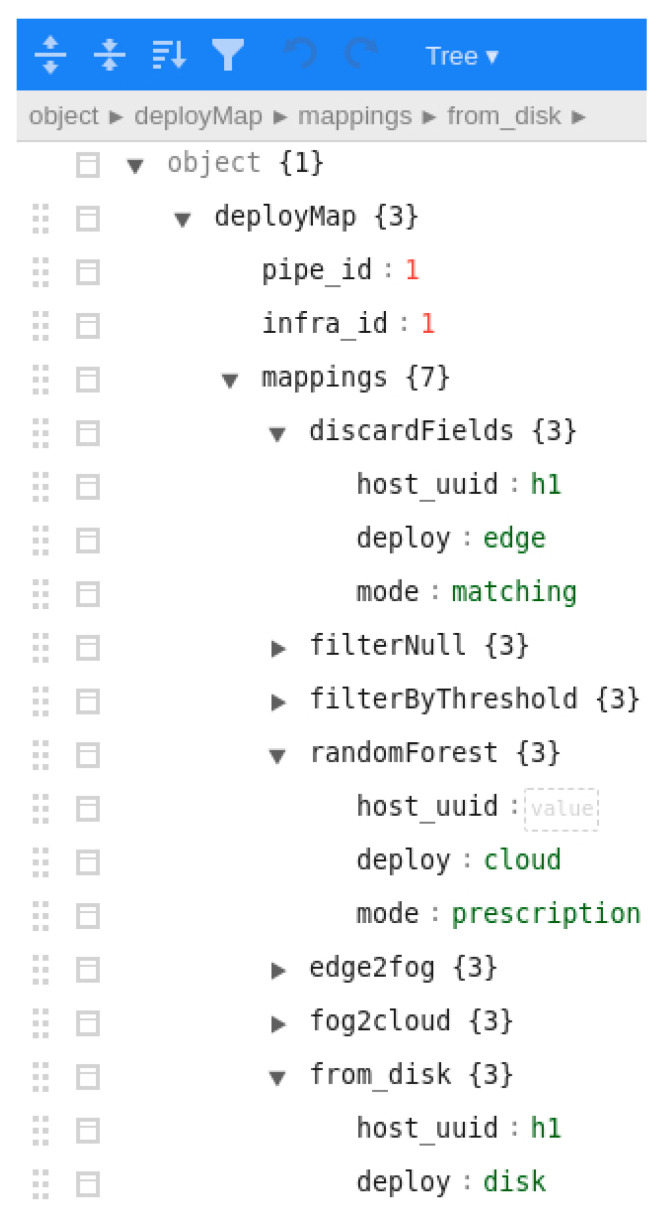
Screenshot of the client when editing the deploy map document.

**Figure 13 sensors-22-04425-f013:**
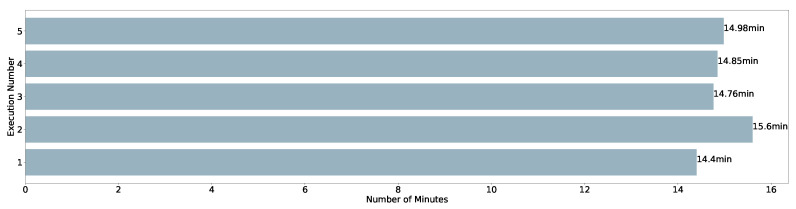
Time spent to execute Pangea over different executions.

**Figure 14 sensors-22-04425-f014:**
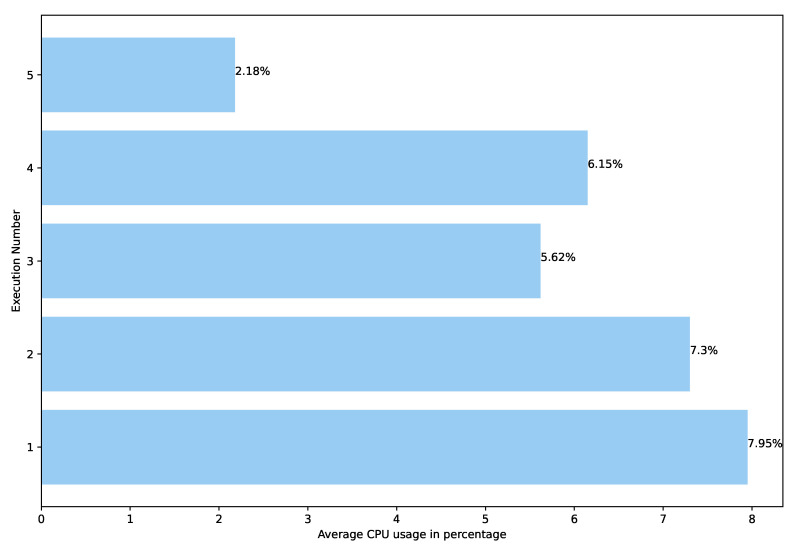
RAM consumed by Pangea over different executions.

**Figure 15 sensors-22-04425-f015:**
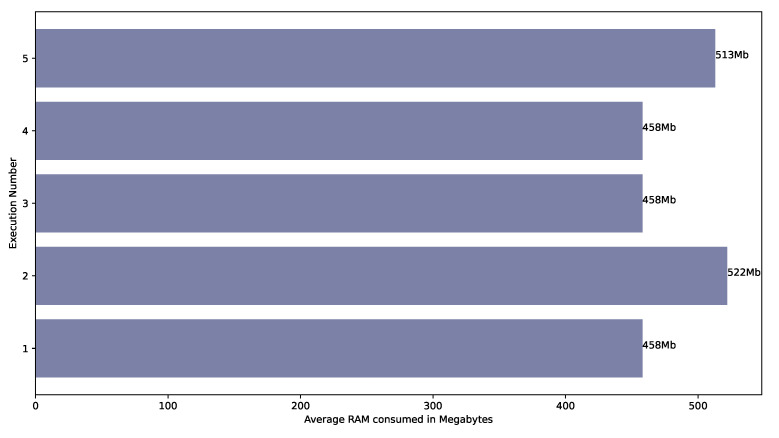
CPU usage by Pangea over different executions.

**Figure 16 sensors-22-04425-f016:**
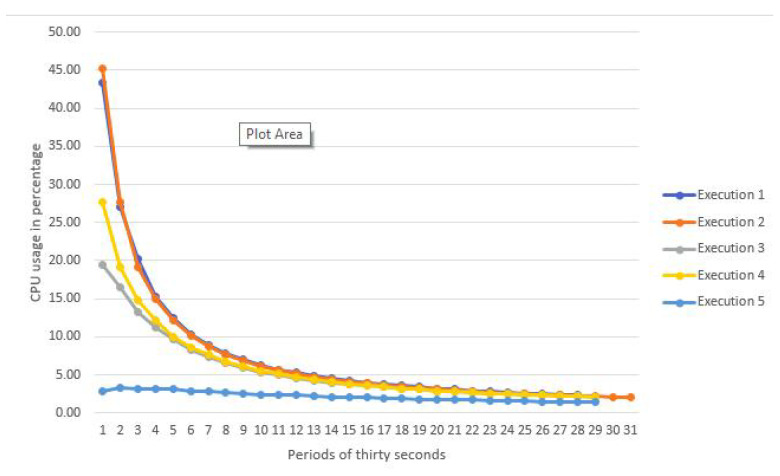
Evolution of the CPU usage over a period of time.

**Table 1 sensors-22-04425-t001:** Comparison among the key features of the main provisioning tools (extracted from [21]). symbol means that the technology satisfies the requirement.

Metrics	Chef	Puppet	Ansible	Salt
Availability	√	√	√	√
Ease of setup	Not very easy	Not very easy	Easy	Not very easy
Management	Not very easy	Not very easy	Easy	Easy
Scalability	Highly scalable	Highly scalable	Highly scalable	Highly scalable
Conf. Language	DSL (Ruby)	DSL (Puppet DSL)	YAML (Python)	YAML (Python)
Pricing (up to 100 nodes)	USD 13,700	USD 11,200–19,900	USD 10,000	USD 15,000

**Table 2 sensors-22-04425-t002:** Related work compared against the main functionalities of Pangea. √ symbol means that the technology satisfies the requirement.

Technology	Generate Infrastructure	Provision Machines	Deployment in Distinct Machines	Edge, Fog, Cloud Support
MLFlow	-	-	-	√
Scanflow	-	-	-	√
Clipper	-	-	-	√
ML.NET	-	-	-	√
ZOO	-	-	-	√
PyCaret	-	-	-	-
Seldom	-	-	-	-
ECO	-	-	√	√
STRATUM	√	√	√	√
TFX	-	-	√	√
TorchServe	-	-	√	√
Kubeflow	-	-	√	-
Pangea	√	√	√	√

**Table 3 sensors-22-04425-t003:** Vagrant machines’ configuration for simulating the necessary nodes.

Name	IP	Node	CPUs	RAM (Mb)
node1	192.168.0.11	EDGE	1	512
node2	192.168.0.12	FOG	2	4096
nq1	192.168.0.21	MQTT	2	2048
nq2	192.168.0.22	Kafka	2	4096

## Data Availability

Not applicable.

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
