# Peer review of "Pangea: An MLOps Tool for Automatically Generating Infrastructure and Deploying Analytic Pipelines in Edge, Fog and Cloud Layers"

_sensors, 2022, doi:10.3390/s22124425_

Round 1

Reviewer 1 Report

The article proposes an automated ML deployment system called PANGEA. PANGEA is focused on the packaging, deployment, and fulfillment phases of the ML lifecycle, allowing professionals unfamiliar with data engineering to deploy ML models. 

This work does not make a tangible scientific contribution to the community. Pangea looks more like a tool that can be used to easily deploy ML projects. 

The motivation of the work is not clear or it is irrelevant.

A few more general comments:

  • English must be revised.
  • References are basically consisted of online sites. Recent and relevant articles must be included.
  • The article should be reorganized, for example: motivation should be in the introduction, background and related works should be in different Sections.
  • The article has many irrelevant discussions, which makes it long and promiscuous causing reader disinterest.
  • Some acronyms were not explained.
  • There are missed references.

Author Response

Dear reviewer,

We would like to thank you for the constructive comments and useful suggestions on our submitted manuscript. We have carefully addressed all such comments, which have been of great help for improving the coverage, completeness, and depth of discussion of our work.

This response letter details all changes made to the original manuscript, which are highlighted with tracking changes in the attached version of the manuscript.

We sincerely hope that you find in this revision the quality required for its publication in the Special Issue "Recent Advances in Big Data and Cloud Computing" in the Sensors MDPI journal. In any case, we remain at your disposal for any further related matter.

Sincerely yours,

Raúl Miñón (on behalf of all the co-authors)

-------------

Reviewer #1:

Comment 1.1: “This work does not make a tangible scientific contribution to the community. Pangea looks more like a tool that can be used to easily deploy ML projects.”

Response 1.1: We have introduced changes all over the paper to better highlight the contributions of the research.

Comment 1.2: “The motivation of the work is not clear or it is irrelevant.”

Response 1.2: Some modifications have been done in this section to improve it.

Comment 1.3: “English must be revised.”

Response 1.3: English has been carefully reviewed and several errors have been repaired. In addition, several sentences have been modified to simplify complex ones.

Comment 1.4: “References are basically consisted of online sites. Recent and relevant articles must be included.”

Response 1.4: The first two references in the introduction have been substituted for more updated scientific ones. The related work has been enhanced including three new technologies based on scientific papers (Scanflow, Clipper and TorchServe). Salt and Ansible online references have been changed by book references. In addition, the TensorFlow eXtended and Puppet references have been modified by scientific ones Finally, scientific references have been included for the DevOps and Infrastructure as Code concepts.

Comment 1.5: “The article should be reorganized, for example: motivation should be in the introduction, background and related works should be in different Sections.”

Response 1.5: The introduction has been divided into four subsections including the motivation example. Moreover, the background and related work have been separated into two sections.

Comment 1.6: “The article has many irrelevant discussions, which makes it long and promiscuous causing reader disinterest.”

Response 1.6: Some redundant texts or certain irrelevant discussions have been removed to deal with this issue. Moreover, previous sections 5.1 and 5.3 have been merged into section 4.1 (previously 5.1).

Comment 1.7: “Some acronyms were not explained.”

Response 1.7: The missing acronyms have been explained.

Comment 1.8: “There are missed references.”

Response 1.8: Some references targeting to tables and figures have been repaired. In addition, all the scientific references have been checked and substituted when necessary.

Reviewer 2 Report

please remove the list code from the paper.

  This paper designs a tool named Pangea, which is used to manage some IT infrastructure in key fields and realize its automatic analysis process in this field. Some problems in the implementation process are discussed. At present, the tool only considers the packaging, deployment and service stage of the trained pipeline, and is not designed as the machine learning training, testing and verification stage. In addition, the compatibility of the tool with new methods of providing models will be studied.

Recommendations:

(1) For the specific operation process of Pangea tool and the purpose of each step, a simple description can be written uniformly;

(2) Before comparing your technologies in the same field, you can introduce other technologies in more detail;

(3) Some simple comments can be made in the code part, which is more convenient to view.

Author Response

RESPONSE LETTER

Title: “Pangea: an MLOps Tool for Automatically Generating Infrastructure and Deploying Analytic Pipelines in Edge, Fog and Cloud Layers”

Authors: Raúl Miñón, Josu Diaz-de-Arcaya, Ana I. Torre-Bastida and Philipp Hartlieb

Dear reviewer,

We would like to thank you for the constructive comments and useful suggestions on our submitted manuscript. We have carefully addressed all such comments, which have been of great help for improving the coverage, completeness, and depth of discussion of our work.

This response letter details all changes made to the original manuscript, which are highlighted with tracking changes in the attached version of the manuscript.

We sincerely hope that you find in this revision the quality required for its publication in the Special Issue "Recent Advances in Big Data and Cloud Computing" in the Sensors MDPI journal. In any case, we remain at your disposal for any further related matter.

Sincerely yours,

Raúl Miñón (on behalf of all the co-authors)

-------------

Reviewer #2:

Comment 2.1: “For the specific operation process of Pangea tool and the purpose of each step, a simple description can be written uniformly”

Response 2.1: Previous sections 5.1 and 5.3 have been merged into section 4.1 (previously 5.1). In addition, some redundant texts have been removed.

Comment 2.2: “Before comparing your technologies in the same field, you can introduce other technologies in more detail;”

Response 2.2: We admit that the Related Work was merged with some background works. Consequently, a background section has been included before the related work.

Comment 2.3: “Some simple comments can be made in the code part, which is more convenient to view.”

Response 2.3: Code excerpts have been enriched with some comments.

Reviewer 3 Report

Dear Authors,

I appreciate your work and the proposal made. However, some further clarifications would be needed.

A first comment is related to the figures included within the paper. My recommendation is to insert (where is the case) info if the figure represents an own design of authors or if there are describing adapted representations of some other authors/approaches.

Regarding the proposed structure, I would have suggested to include a logical thread or a workflow that would simplify the understanding of the design of the article content, proposed by the authors.

Even though the structure of the paper is not exactly the recommended one, the approach proposed by the authors can be accepted. In this regard I would mention section 2 entitled Motivation Example which brings arguments and explanations regarding the applicability of the proposed tool PANGEA. Related to this issue I have a question. You mentioned that Pangea can be used in the mining industry. Your initial thoughts aimed at applying the tool in this industry? Why? This is because of some specific features such as: risk of danger of rock bursts, seismic events etc? This tool could be also used, for instance, in another extractive industry such as oil industry (upstream sector)? 

At what level of applicability can this Pangea tool be used?  Because I mentioned this dilemma, a lack of paper refers to the limitations (actually not mentioning them).

In order to underline better the findings within the entire paper I would like to make few suggestions to be inserted in the last part of the paper:

- please highlights the main contribution of the authors to the domain;

- explain which are the main advantages and disadvantages, as well, for this particular tool (offering a better image for what propose the tool and its performance);

- describe how difficult is to customize the web client solution in order to offer a wide applicability of the proposed tool.

Author Response

Dear reviewer,

We would like to thank you for the constructive comments and useful suggestions on our submitted manuscript. We have carefully addressed all such comments, which have been of great help for improving the coverage, completeness, and depth of discussion of our work.

This response letter details all changes made to the original manuscript, which are highlighted with tracking changes in the attached version of the manuscript.

We sincerely hope that you find in this revision the quality required for its publication in the Special Issue "Recent Advances in Big Data and Cloud Computing" in the Sensors MDPI journal. In any case, we remain at your disposal for any further related matter.

Sincerely yours,

Raúl Miñón (on behalf of all the co-authors)

-------------

Reviewer #3:

Comment 3.1: “A first comment is related to the figures included within the paper. My recommendation is to insert (where is the case) info if the figure represents an own design of authors or if there are describing adapted representations of some other authors/approaches”

Response 3.1: We appreciate this proposal. However, we have not included any specific information since all the pictures are designed by our own. On the other hand, Table 1 was extracted from other source which was already mentioned.

Comment 3.2: “Regarding the proposed structure, I would have suggested to include a logical thread or a workflow that would simplify the understanding of the design of the article content, proposed by the authors.”

Response 3.2: We have restructured various parts of the article to simplify the structure and the reading. Moreover, in Section 1.4 we have extended the structure description to better guide the reader.

Comment 3.3: “Even though the structure of the paper is not exactly the recommended one, the approach proposed by the authors can be accepted. In this regard I would mention section 2 entitled Motivation Example which brings arguments and explanations regarding the applicability of the proposed tool PANGEA. Related to this issue I have a question. You mentioned that Pangea can be used in the mining industry. Your initial thoughts aimed at applying the tool in this industry? Why? This is because of some specific features such as: risk of danger of rock bursts, seismic events etc? This tool could be also used, for instance, in another extractive industry such as oil industry (upstream sector)?”

Response 3.3: The introduction has been divided into four subsections including the motivation example. Moreover, the background and related work have been separated into two sections. Finally, previous sections 5.1 and 5.3 have been merged into section 4.1 (previously 5.1). In addition, some redundant texts have been removed.

In the motivation example section, it has been clarified that Pangea is not restricted to any specific domain. Any domain requiring the deployment of machine learning models or analytical pipelines could benefit from the use of Pangea.

Comment 3.4: “At what level of applicability can this Pangea tool be used?  Because I mentioned this dilemma, a lack of paper refers to the limitations (actually not mentioning them).”

Response 3.4: In Subsection Motivation Example (currently Subsection 1.3) some lines have been mentioned in this direction and a paper where PADL (the language used by Pangea to define analytical pipelines) is proposed to solve two use cases in the flood control and waste management domains.

Comment 3.5: “In order to underline better the findings within the entire paper I would like to make few suggestions to be inserted in the last part of the paper:”

  • please highlights the main contribution of the authors to the domain
  • explain which are the main advantages and disadvantages, as well, for this particular tool (offering a better image for what propose the tool and its performance);
  • describe how difficult is to customize the web client solution in order to offer a wide applicability of the proposed tool.

Comment 3.5: We have included the first two suggestions in the conclusions section. The last one has not been addressed since we consider that there is neither any special novelty nor specific problems beyond the web development nuances when building the web client.

Round 2

Reviewer 1 Report

No further comments.